# Formal [3+2] Cycloaddition Reactions of Electron-Rich Aryl Epoxides with Alkenes under Lewis Acid Catalysis Affording Tetrasubstituted Tetrahydrofurans

**DOI:** 10.3390/molecules25030692

**Published:** 2020-02-06

**Authors:** Víctor E. Macías-Villamizar, Luís Cuca-Suárez, Santiago Rodríguez, Florenci V. González

**Affiliations:** 1Departament de Química Inorgànica i Orgànica, Universitat Jaume I, 12080 Castelló, Spain; vmacias69@gmail.com (V.E.M.-V.); rodrigue@uji.es (S.R.); 2Departamento de Química, Universidad Nacional de Colombia, Bogotá D.C. 111321, Colombia; lecucas@unal.edu.co

**Keywords:** lignans, epoxides, cycloaddition

## Abstract

We report on the regio- and stereoselective synthesis of tetrahydrofurans by reaction between epoxides and alkenes in the presence of a Lewis acid. This is an unprecedented formal [3+2] cycloaddition reaction between an epoxide and an alkene. The chemical reaction represents a very concise synthesis of tetrahydrofurans from accessible starting compounds.

## 1. Introduction

Diaryl dimethyltetrahydrofurans lignans (Figure 1) are an important group of natural products displaying anti-apoptotic, anticancer, and anti-inflammatory properties [1,2,3] and interesting new examples have been found during the last few years [4]. Reported synthetic approaches to these compounds are usually step intensive, including functional group manipulations [5,6].

Formal [3+2] cycloadditions represent an interesting synthetic approach for the preparation of tetrahydrofurans. Some interesting examples have been recently reported, for example, the combination of allylic silanes and aldehydes in the presence of a Lewis acid [7], the reaction between ketenes and vinyl cyclopropanes under palladium catalysis [8] or the cycloaddition between a palladium-oxyallyl complex and a diene [9]. On the other hand, epoxides are versatile intermediates in organic synthesis. Ring opening of epoxides by the use of an acid catalyst are attractive reactions which require effective ways to control the regioselectivity [10]. Disubstituted tetrahydrofurans have been prepared by reaction between epoxides and alkenes [11]. Cascade cyclizations of polyepoxides stand as one of the most beautiful and efficient reactions in organic synthesis [12]. In some of these intramolecular reactions, an alkene is the initiating nucleophile that attacks the first epoxide which will trigger the process.

Inspired by these transformations, we envisioned the combination of an epoxide and an alkene under acidic conditions as a straightforward synthetic approach to tetrasubstituted tetrahydrofurans (Figure 1). The concept of this reaction revolves around the idea of the epoxide as an oxygenated cationic synthon and the alkene as a 1,2-dipole to afford a tetrasubstituted tetrahydrofuran (Figure 1). 

We report herein the preparation of tetrahydrofurans by combination of an epoxide and an alkene in the presence of a Lewis acid. 

## 2. Results

We began our study by combining one equivalent of methyl isoeugenol and one equivalent of its corresponding epoxide with boron trifluoride (30 mol %) in dichloromethane. These conditions were chosen based on previous results reported by us [10]. We were pleased to see that the reaction afforded the desired compound as a 7:3 mixture of stereoisomers **1a** and **2a** (Scheme 1), although the chemical yield was not satisfactory. Under these conditions, undesired byproducts were also formed, including cyclobutane resulting from self-condensation of the alkene. In order to increase the yield, the reaction was performed using 2 equivalents of the alkene and 1 equivalent of the epoxide but this resulted in a lower yield. Furthermore, if more equivalents of the epoxide than the alkene were added, the chemical yield did not improve. Higher chemical yield and similar stereoselection were obtained by using aluminum chloride as a Lewis acid instead of boron trifluoride (Scheme 1). Aluminum chloride is a cheap Lewis acid and it was therefore advantageous that it worked well in this purpose.

To study the scope of the reaction, various β-methyl styrenes and epoxides were subjected to the optimized reaction conditions (Scheme 2). We started by combining methyl isoeugenol oxide with less electron-rich alkenes. Methyl isoeugenol oxide was reacted with β-methyl styrene affording compound **1b** in similar chemical yield and stereo-selectivity as **1a** (Scheme 2). However, the reaction with styrene afforded an equimolecular mixture of three isomeric compounds **1d** (Scheme 2). Comparison of the selectivity of **1b** and **1d** denotes that the substituent at the beta position of the alkene is important for stereocontrol. We then assayed the reaction with epoxides derived from less electron-rich alkenes. When stilbene oxide was combined with methyl isoeugenol, tetrahydrofuran **1c** was obtained at very low yield (Scheme 2), affording instead 2,2-diphenylacetaldehyde as a main product. This compound results from Meinwald rearrangement of starting stilbene oxide under acid catalysis. The reaction between styrene oxide and methyl isoeugenol oxide gave a complex mixture of compounds, but no tetrahydrofuran could be isolated. When other electron-rich alkenes different from methyl isoeugenol are used, the cycloaddition process then takes place. Tetrahydrofurans **1e** and **1f** derived from the combination of methyl isoeugenol oxide with α-asarone or isoeugenol were prepared. The *anti/anti/syn* isomer was the major one in both cases. In addition, compound **1g** containing a carboxylic ester was obtained by reaction between methyl 3-arylglycidate and methyl isoeugenol. The selectivity of this reaction is similar to the one obtained when using methyl isoeugenol oxide (Scheme 2).

For all cases except for compound **1d**, the main stereoisomer could be isolated by chromatography. Further investigation of the reaction with other substitution patterns is currently underway.

## 3. Discussion

Unambiguously established 2,3-*anti*, 3,4-*anti*, 4,5-*syn* configuration of the natural product magnosalicin [13] enabled us to use its NMR data as our standard for tentative stereochemical assignment. Chemical shifts and coupling constants of **1a**, **1b**, **1e**, **1f** and **1g** were very similar to magnosalicin (Figure 2). In addition, the NOE data (see Appendix A) denote the stereochemistry to be *anti/anti/syn* (Figure 2). The structure of the minor isomers is *anti/anti/anti* as assigned by comparison of ^1^H NMR spectrum with already reported *epi*-magnosalicin: as compared to **2a**, a downfield shift of the C-2 CH_3_ signal, and an upfield shift of the H-3 signal are observed. Interestingly, relative 2,4-*syn* configuration has also been reported for the preparation of disubstituted tetrahydrofurans under similar conditions [11]. 

Chemical correlation of compound **1g** with **1a** was done through a three-step sequence. The first step is the reduction of the ester group, then activation of the resulting alcohol as a *p*-toluenesulfonate ester and, finally, conversion to methyl (Scheme 3).

In order to explain the reaction between an alkene and an epoxide a two-step mechanism is proposed. Initial attack of β-carbon of alkene to α-carbon of activated epoxide would afford a benzylic carbocation, stabilized by para-quinone methide structures, which upon ring closure would give the tetrahydrofuran (Scheme 4). Stereocontrol would result from the first step through a less strained transition state resulting from the combination of both planar reactants and subsequent cyclization through attack of the oxygen atom on the less hindered face of the carbocation (Scheme 4).

## 4. Materials and Methods 

### 4.1. General Information

Unless otherwise specified, all reactions were carried out under nitrogen atmosphere with magnetic stirring. All solvents and reagents were obtained from commercial sources and were purified according to standard procedures before use. ^1^H and ^13^C NMR spectra were measured in CDCl_3_ (^1^H, 7.24 ppm; ^13^C 77.0 ppm) solution at 30 °C on a 300 or 400 MHz NMR spectrometer (Bruker, Rheinstetten, Germany), Mass spectra (Waters, Milford, MA, USA) were measured in a QTOF I (quadrupole–hexapole–TOF) mass spectrometer with an orthogonal Z-spray–electrospray interface. EM Science Silica Gel 60 was used for column chromatography while Thin Layer Chromatograpgy (TLC) (Merck, Darmstadt, Germany) was performed with precoated plates (Kieselgel 60, F_254_, 0.25 mm). 

### 4.2. Preparation of Epoxides 2-(3,4-dimethoxyphenyl)-3-methyloxirane and 2,3-diphenyloxirane

A stirred solution of the alkene (5.61 mmol) in dichloromethane (20 mL) at 0–5 °C (ice-bath) was treated with sodium carbonate (10% aqueous solution) (40 mL). Then, a solution of *m*-chloroperbenzoic acid (12.4 mmol) in dichloromethane (20 mL) was added dropwise to the previous mixture. The resulting mixture was stirred for 20 min, then was poured into a separatory funnel. The organic layer was separated and washed sequentially with sodium carbonate (10% aqueous solution) (5 × 25 mL) and dried over Na_2_SO_4_. Finally, the solvent was evaporated and the residue was purified by column chromatography (silicagel, n-hexane/ethyl acetate; 9:1).

### 4.3. Preparation of Methyl 3-(3,4-dimethoxyphenyl)oxirane-2-carboxylate

Sodium (62 g) was added to dry methanol (900 mL) and the resulting solution was cold to −10 °C. Then, a solution of veratraldehyde (1.8 mol) and methyl chloroacetate (293 g) in methanol (mL) was added dropwise over a period of 3 h with vigorous stirring. The resulting mixture was stirred for 2 h at −5 °C and then for 3 h at room temperature. The mixture was poured into a flask containing a mixture of water, ice and acetic acid (20 mL) (total volume of the mixture was 3.5 L). The desired compound precipitated as a white powder, and was filtered, washed with cold water and dried. The solid was recrystallized from methanol to afford a white solid (65–66 °C).

### 4.4. General Experimental Procedure for the Preparation of Tetrahydrofurans

To an ice-bath cold solution of alkene (0.56 mmol) in dichloromethane (1 mL) the corresponding epoxide was added (0.56 mmol, 1 equiv), then aluminum chloride (0.168 mmol, 0.3 equiv) was added. The resulting mixture was stirred cold with an ice-bath for 30 min. An indicative color change was observed after this time. Water (10 mL) was then added followed by extraction with dichloromethane (3 × 20 mL). Combined organic layers were dried (Na_2_SO_4_), filtered and concentrated to afford an oil which was purified through silica gel column.

**(2*R*,3*R*,4*S*,5*R*)-2,4-bis(3,4-dimethoxyphenyl)-3,5-dimethyltetrahydrofuran 1a and (2*R*,3*R*,4*S*,5*S*)-2,4-bis(3,4-dimethoxyphenyl)-3,5-dimethyltetrahydrofuran 2a.** (156 mg, 75%) (hexane/ethyl acetate, 75/25 and 60/40). **1a**: ^1^H NMR (300 MHz, CDCl_3_) δ 7.00 (m, 2H), 6.86 (m, 2H), 6.72 (m, 2H), 4.46 (m, 1H), 3.92 (s, 3H), 3.87 (s, 3H), 3.18 (dd, *J* = 9.6, 8.5 Hz, 1H), 2.32 (m, 1H), 1.05 (d, *J* = 6.5 Hz, 3H), 0.97 (d, *J* = 6.5 Hz, 3H) ppm; ^13^C NMR (75 MHz, CDCl_3_) d 149.2, 149.0, 148.8, 148.0, 134.3, 132.2, 120.8, 118.7, 112.4, 111.3, 111.2, 109.8, 87.7, 77.6, 77.4, 56.8, 56.2, 56.1, 56.0, 46.8, 19.1, 15.3 ppm. HRMS (ESI) calcd for C_22_H_28_O_2_Na (M + Na^+^) 395.1834, found 395.1831). **2a**: ^1^H NMR (300 MHz, CDCl_3_) δ 6.89 (m, 2H), 6.76 (m, 2H), 6.64 (m, 2H), 4.49 (d, *J* = 9.4 Hz, 1H), 4.22 (dq, *J* = 12.0, 6.0 Hz, 1H), 3.85 (s, 3H), 3.81 (s, 3H), 3.80 (s, 3H), 3.79 (s, 3H), 2.46 (dd, *J* = 11.0, 9.5 Hz, 1H), 2.32 (m, 1H), 1.22 (d, *J* = 6.0 Hz, 3H), 0.86 (d, *J* = 6.5 Hz, 5H) ppm.

**(2*R*,3*S*,4*R*,5*R*)-3-(3,4-dimethoxyphenyl)-2,4-dimethyl-5-phenyltetrahydrofuran 1b and (2*R*,3*S*,4*R*,5*S*)-3-(3,4-dimethoxyphenyl)-2,4-dimethyl-5-phenyltetrahydrofuran 2b.** (108 mg, 62%) (hexanes/ethyl acetate, 75/25 and 60/40). ^1^H NMR (300 MHz, CDCl_3_) d 7.29–7.45 (m, 5H), 6.68–6.82 (m, 3H), 4.46 (m, 1H), 4.49 (d, *J* = 9.1 Hz, 1H), 3.86 (s, 3H), 3.84 (s, 3H), 3.18 (dd, *J* = 9.6, 8.2 Hz, 1H), 2.31 (m, 1H), 1.05 (d, *J* = 6.5 Hz, 3H), 0.97 (d, *J* = 6.5 Hz, 3H) ppm; ^13^C NMR (100 MHz, CDCl_3_) d 148.8, 147.5, 141.5, 131.9, 128.2, 127.6, 126.0, 120.6, 111.9, 111.0, 87.5, 56.6, 55.7, 47.0, 18.8, 15.0 ppm. HRMS (ESI) calcd for C_20_H_24_O_3_Na (M + Na^+^) 335.1623, found 335.1625. **2b**: ^1^H NMR (400 MHz, CDCl_3_) δ 7.39–7.20 (m, 5H), 6.62–6.77 (m, 3H), 4.54 (d, *J* = 9.4 Hz, 1H), 4.29–4.17 (m, 1H), 3.80 (s, 3H), 3.79 (s, 3H), 2.47 (dd, *J* = 11.1, 9.5 Hz, 1H), 2.18 (m, 1H), 1.22 (d, *J* = 6.0 Hz, 3H), 0.87 (d, *J* = 6.5 Hz, 3H) ppm.

**(2*S*,3*R*,4*S*,5*R*)-2-(3,4-dimethoxyphenyl)-3-methyl-4,5-diphenyltetrahydrofuran 1c.** (31 mg, 15%) (hexanes/ethyl acetate, 75/25 and 60/40). ^1^H NMR (500 MHz, CDCl_3_) d 6.56–6.79 (m, 6H), 6.90–6.93 (m, 2H), 7.19–7.33 (m, 5H), 4.04 (d, *J* = 9.6 Hz,1H), 3.88 (d, *J* = 9.2 Hz, 1H), 3.82 (s, 3H), 3.70–3.79 (m, 1H), 3.75 (s, 3H), 2.13 (m, 1H), 0.96 (d, *J* = 6.4 Hz, 3H) ppm; ^13^C NMR (100 MHz, CDCl_3_) d 149.1, 147.7, 143.2, 139.9, 138.5, 137.5, 129.9, 129.3, 128.9, 127.2, 126.2, 126.1, 122.1, 112.2, 111.0, 78.0, 77.2, 56.0, 55.9, 53.7, 43.3, 16.0 ppm. HRMS (ESI) calcd for C_20_H_24_O_3_Na (M + Na^+^) 397.1780, found 397.1779.

**(2*R*,3*S*,5*R*)-3-(3,4-dimethoxyphenyl)-2-methyl-5-phenyltetrahydrofuran 1d.** (95 mg, 57%) (hexanes/ethyl acetate, 75/25 and 70/30). ^1^H NMR (300 MHz, CDCl_3_) δ 7.63–7.17 (m, 8H), 6.96–6.62 (m, 5H), 5.23 (dd, *J* = 12.0, 5.6 Hz, 1H), 5.07 (dd, *J* = 9.5, 6.7 Hz, 1H), 4.52–4.39 (m, 1H), 4.24 (dq, *J* = 9.1, 6.0 Hz, 1H), 4.12 (dq, *J* = 8.8, 6.0 Hz, 1H), 3.62 (dd, *J* = 15.8, 7.8 Hz, 1H), 3.06 (ddd, *J* = 11.6, 9.2, 6.8 Hz, 1H), 2.94 (q, *J* = 8.9 Hz, 1H), 2.78 (dd, *J* = 12.5, 6.5 Hz, 1H), 2.74 (dd, *J* = 12.6, 2.2 Hz, 1H), 2.60 (t, *J* = 8.7 Hz, 1H), 2.56 (t, *J* = 8.7 Hz, 1H), 2.40 (dd, *J* = 8.8, 5.5 Hz, 1H), 2.35 (dd, *J* = 8.8, 5.4 Hz, 1H), 2.18 (dt, *J* = 22.1, 11.1 Hz, 1H), 2.25 (m, 1H), 1.40 (d, *J* = 6.0 Hz, 2H), 1.36 (d, *J* = 6.0 Hz, 1H), 1.00 (d, *J* = 6.4 Hz, 1H); ^13^C NMR (75 MHz, CDCl_3_) d 149.1, 149.1, 148.7, 148.0, 147.9, 147.6, 143.8, 143.6, 142.7, 128.5, 128.4, 128.4, 125.8, 125.7, 125.6, 119.6, 119.5, 111.7, 111.4, 111.4, 110.9, 110.8, 110.7, 82.8, 82.3, 80.1, 79.9, 79.8, 78.5, 55.9, 55.9, 55.8, 54.1, 51.8, 48.6, 44.7, 43.8, 41.1, 19.6, 19.1, 18.0 ppm. HRMS (ESI) calcd for C_20_H_24_O_3_Na (M + Na^+^) 321.1468, found 321.1468.

**(2*R*,3*S*,4*R*,5*R*)-3-(3,4-dimethoxyphenyl)-2,4-dimethyl-5-(2,4,5 trimethoxyphenyl)tetrahydrofuran 1e.** (92 mg, 41%) (hexanes/ethyl acetate, 75/25 and 60/40). ^1^H NMR (300 MHz, CDCl_3_) d 7.07 (s, 1H), 6.73 (s, 1H), 6.62–6.67 (m, 2H), 6.46 (s, 1H), 4.86 (d, *J* = 9.0 Hz,1H), 4.36 (dq, *J* = 8.2, 6.5 Hz, 1H), 3.82 (s, 3H), 3.80 (s, 3H), 3.79 (s, 3H), 3.78 (s, 3H), 3.73 (s, 3H), 3.09 (dd, *J* = 9.1, 8.4 Hz, 1H), 2.31 (m, 1H), 0.98 (d, *J* = 6.6 Hz, 3H), 0.89 (d, *J* = 6.5 Hz, 3H) ppm; ^13^C NMR (75.5 MHz, CDCl_3_) d 151.8, 149.0, 148.8, 147.7, 143.4, 132.4, 97.7, 81.0, 77.2, 56.9, 56.7, 56.5, 56.2, 55.9, 55.9, 46.5, 18.8, 15.4 ppm. HRMS (ESI) calcd for C_23_H_30_O_6_Na (M + Na^+^) 425.1940, found 425.1939.

**4-((2*R*,3*R*,5*R*)-4-(3,4-dimethoxyphenyl)-3,5-dimethyltetrahydrofuran-2-yl)-2-methoxyphenol 1f and 4-((2*R*,3*R*,5*S*)-4-(3,4-dimethoxyphenyl)-3,5-dimethyltetrahydrofuran-2-yl)-2-methoxyphenol 2f.** (70 mg, 35%) (hexanes/ethyl acetate, 75/25 and 60/40). **1f**: ^1^H NMR (300 MHz, CDCl_3_) δ 6.91 (s, 1H), 6.85 (s, 1H), 6.62–6.78 (m, 3H), 5.57 (br s, 1H), 4.36 (dd, *J* = 8.1, 6.5 Hz, 1H), 4.35 (d, *J* = 9.2 Hz,1H), 3.84 (s, 3H), 3.80 (s, 3H), 3.79 (s, 3H), 3.10 (dd, *J* = 9.9, 8.2 Hz, 1H), 2.22 (m, 1H), 0.96 (d, *J* = 6.5 Hz, 3H), 0.89 (d, *J* = 6.5 Hz, 3H) ppm; ^13^C NMR (75.5 MHz, CDCl_3_) d 148.8, 147.8, 146.6, 145.2, 133.4, 132.0, 120.6, 119.3, 114.2, 112.2, 111.1, 108.9, 87.7, 56.5, 55.9, 55.9, 55.9, 46.6, 18.9, 15.1 ppm. HRMS (ESI) calcd for C_21_H_26_O_5_Na (M + Na^+^) 381.1678, found 381.1679. **2f**: ^1^H NMR (300 MHz, CDCl_3_) δ 6.89 (s, 1H), 6.84 (s, 1H), 6.62–6.78 (m, 3H), 5.57 (br s, 1H), 4.47 (d, *J* = 9.4 Hz, 1H), 4.2 (dq, *J* = 9.4, 6.0 Hz,1H), 3.86 (s, 3H), 3.82 (s, 3H), 3.80 (s, 3H), 2.46 (dd, *J* = 11.1, 9.4 Hz, 1H), 2.22 (m, 1H), 1.22 (d, *J* = 6.0 Hz, 3H), 0.85 (d, *J* = 6.5 Hz, 3H) ppm.

**Methyl (2*S*,3*S*,4*R*,5*R*)-3,5-bis(3,4-dimethoxyphenyl)-4-methyltetrahydrofuran-2-carboxylate 1g and methyl (2*R*,3*S*,4*R*,5*R*)-3,5-bis(3,4-dimethoxyphenyl)-4-methyltetrahydrofuran-2-carboxylate 2g**. (105 mg, 45%) (hexanes/ethyl acetate, 75/25 and 60/40). **1g**: ^1^H NMR (300MHz, CDCl_3_) δ 7.55 (d, *J* = 1.9 Hz, 1H), 7.10 (m, 1H), 6.88–6.71 (m, 4H), 4.81 (d, *J* = 9.4 Hz, 1H), 4.55 (d, *J* = 9.7 Hz, 1H), 3.97 (s, 3H), 3.90 (s, 3H), 3.87 (s, 3H), 3.86 (s, 3H), 3.84 (s, 3H), 3.43 (dd, *J* = 11.9, 9.4 Hz, 1H), 2.67–2.50 (m, 1H), 0.92 (d, *J* = 6.5 Hz, 3H) ppm; ^13^C NMR (75MHz, CDCl_3_) d 172.6, 149.4, 149.0, 148.5, 133.1, 128.3, 120.4, 119.5, 111.5, 111.2, 110.7, 110.3, 89.2, 80.9, 56.5, 55.8, 51.4, 45.3, 13.0 ppm. HRMS (ESI) calcd for C_23_H_28_O_7_Na (M + Na^+^) 439.1732, found 439.1733. **2g**: ^1^H NMR (300 MHz, CDCl_3_) δ 7.55 (d, *J* = 1.9 Hz, 1H), 6.92 (m 5H), 4.71 (d, *J* = 9.1 Hz, 1H), 4.68 (d, *J* = 8.2 Hz, 1H), 3.97 (s, 3H), 3.92 (s, 3H), 3.89 (s, 3H), 3.87 (s, 3H), 3.86 (s, 3H), 3.16 (dd, *J* = 10.8, 8.6 Hz, 2H), 2.19 (m, 1H), 0.95 (d, *J* = 6.5 Hz, 3H) ppm.

## 5. Conclusions

In summary, tetrasubstituted tetrahydrofurans can be prepared in a highly regio- and diastereoselective fashion when equimolecular amounts of styrenes and styrene oxides are treated with a Lewis acid catalyst, which represents a convenient method of preparing tetrahydrofurans from accessible starting compounds. The reaction opens new possibilities for the preparation of biomedically interesting tetrahydrofurans.

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
