# Peer review of "Formal [3+2] Cycloaddition Reactions of Electron-Rich Aryl Epoxides with Alkenes under Lewis Acid Catalysis Affording Tetrasubstituted Tetrahydrofurans"

_molecules, 2020, doi:10.3390/molecules25030692_

Round 1

Reviewer 1 Report

The article entitled "Formal [3+2] Cycloaddition Reactions of Epoxides with Alkenes under Lewis Acid Catalysis Affording Tetrasubstituted Tetrahydrofurans" by Víctor E. Macías et al. reported a regio- and stereoselective synthesis of tetrahydrofurans derivatives. Reactions were performed by a formal [3+2] cycloaddition reaction between an epoxide and an alkene. The methods, results and discussions were all sufficient and clear.

                Concerning the descriptive aspect of the investigated problem the paper fulfills the goals proposed by the authors. Nevertheless there are some minor questions which should be clarified.

The authors state: “These conditions were chosen based on previous results reported by us[7].”, but none of the authors of the current manuscript is present as the author in the indicated reference (Angle, S. R.; Choi, I. Regioselective and stereoselective synthesis of tetrahydrofurans from a functionalized allylic silane and an aldehyde via formal [3+2]-cycloaddition reaction. Tetrahedron Lett. 2008, 49, 6245-6249.); Since higher chemical yield and similar stereoselection was obtained by using aluminum trichloride as a Lewis acid instead of boron trifluoride, why the author choose anyway boron trifluoride as Lewis acid when represent the reaction mechanism (Scheme 4)? In my opinion is more indicated to use aluminum chloride as Lewis acid when represent the reaction mechanism.

Having in view these considerations I recommend to accept the manuscript in the journal Molecules after minor revisions.

Author Response

Dear reviewer:

thanks for your comments. Please find below the responses to your comments.

The article entitled "Formal [3+2] Cycloaddition Reactions of Epoxides with Alkenes under Lewis Acid Catalysis Affording Tetrasubstituted Tetrahydrofurans" by Víctor E. Macías et al. reported a regio- and stereoselective synthesis of tetrahydrofurans derivatives. Reactions were performed by a formal [3+2] cycloaddition reaction between an epoxide and an alkene. The methods, results and discussions were all sufficient and clear.

                Concerning the descriptive aspect of the investigated problem the paper fulfills the goals proposed by the authors. Nevertheless there are some minor questions which should be clarified.

The authors state: “These conditions were chosen based on previous results reported by us[7].”, but none of the authors of the current manuscript is present as the author in the indicated reference (Angle, S. R.; Choi, I. Regioselective and stereoselective synthesis of tetrahydrofurans from a functionalized allylic silane and an aldehyde via formal [3+2]-cycloaddition reaction. Tetrahedron Lett. 200849, 6245-6249.);

Response: The reviewer was right, there was a mistake: ref. 10 instead of ref. 7 should be indicated. It has been changed.

Since higher chemical yield and similar stereoselection was obtained by using aluminum trichloride as a Lewis acid instead of boron trifluoride, why the author choose anyway boron trifluoride as Lewis acid when represent the reaction mechanism (Scheme 4)? In my opinion is more indicated to use aluminum chloride as Lewis acid when represent the reaction mechanism.

Response: It has been changed as suggested by the reviewer.

Having in view these considerations I recommend to accept the manuscript in the journal Molecules after minor revisions.

Thank-you

Reviewer 2 Report

The present manuscript describes a method for the preparation of tetrasubstituted terahydrofurans upon 3+2 addition between epoxides and alkenes catalyzed by a Lewis acid. Such a synthetic approach was previously proposed in the literature (ref. 11) for the preparation of more simple, disubstituted tetrahydrofurans. The literature method was catalyzed by scandium(III) species in low amounts, whereas in the present method group 13 Lewis acids are employed in high amounts (30%) as promoters. Is there any special reason to turn to these Lewis acids instead of using scandium? Has an optimisation study been carried out concerning the nature and amount of Lewis acid? The authors should clarify this point.

I am also a bit confused concerning the characterization of the obtained product mixtures. Yields in the desired tetrahydrofurans are generally moderate, but only in one case (product 1c) one other byproduct of the reaction is reported. Do the authors have an idea of the fate of the reagents in the other cases? Furthermore, the product is obtained as different stereoisomers, whose nature is described when two stereoisomers are present but not when three stereoisomers are present (products 1c and 1d); which stereoisomers are they? Overall, I find it quite misleading to provide a single code for the mixture of stereoisomer (e.g. "1a"), and then to report the characterization data of one single stereoisomer of them in the Experimental Section. The authors should:

1) Code each stereoisomer separately and define its stereochemistry;

2) Clarify whether the different stereoisomers are obtained as a mixture or if they are separated and isolated in pure form:

3) Report the available characterization data of all steroisomers.

Should the authors cope with the points listed above, the present manuscript, although quite concise, could become suitable for publication in "Molecules"

Author Response

Dear reviewer:

thanks for your comments. Please find below the responses to your comments.

The present manuscript describes a method for the preparation of tetrasubstituted terahydrofurans upon 3+2 addition between epoxides and alkenes catalyzed by a Lewis acid. Such a synthetic approach was previously proposed in the literature (ref. 11) for the preparation of more simple, disubstituted tetrahydrofurans. The literature method was catalyzed by scandium(III) species in low amounts, whereas in the present method group 13 Lewis acids are employed in high amounts (30%) as promoters. Is there any special reason to turn to these Lewis acids instead of using scandium? Has an optimisation study been carried out concerning the nature and amount of Lewis acid? The authors should clarify this point.

Response: This work was done before the publication of reference 11. We started this work using BF3·Et2O as a catalyst but then changed to cheaper Lewis acid AlCl3 which actually worked much better as it is explained in the manuscript. We were glad to get this result and no other Lewis acids were assayed (a sentence explaining this has been added). This work is a collaboration between our group in Spain (Europe) with the group in Colombia. Colombian authors are experts in isolation of natural products from Colombia and we are dedicated to chemical synthesis. This work was conceived as a convenient procedure for the preparation of natural products including cheap materials also in Colombia. The experiments were done with AlCl3 which is an affordable catalyst everywhere as compared to Sc(OTf)3. For example, the price for AlCl3 in Spain (Merck) is 76.4€/KG which is around one thousand times cheaper than Sc(OTf)3 (84.4€/G).

I am also a bit confused concerning the characterization of the obtained product mixtures. Yields in the desired tetrahydrofurans are generally moderate, but only in one case (product 1c) one other byproduct of the reaction is reported. Do the authors have an idea of the fate of the reagents in the other cases?

Response: Besides the aldehyde resulting from Meinwald reaction when preparing 1c, also a cyclobutane was identified as a by-product when preparing tetrahydrofuran 1a (see page 2, first paragraph). For the other reactions, a complex mixture of by-products were obtained which could not be identified.

 Furthermore, the product is obtained as different stereoisomers, whose nature is described when two stereoisomers are present but not when three stereoisomers are present (products 1c and 1d); which stereoisomers are they? Overall, I find it quite misleading to provide a single code for the mixture of stereoisomer (e.g. "1a"), and then to report the characterization data of one single stereoisomer of them in the Experimental Section. The authors should:

1) Code each stereoisomer separately and define its stereochemistry;

Response: Each stereoisomer has been now coded separately as suggested: anti/anti/syn is 1x and anti/anti/anti is 2x. In case of 1c and 1d where three stereoisomers are obtained, they could not be separated and the stereochemistry of the third stereoisomer is not known.

2) Clarify whether the different stereoisomers are obtained as a mixture or if they are separated and isolated in pure form:

Response: A new sentence has been added in page 3: “For all cases except for compounds 1c and 1d, main stereoisomer could be isolated by chromatography.

3) Report the available characterization data of all steroisomers.

Response: NMR spectra for minor stereoisomers 2a, 2b, 2f and 2g are now described in the experimental section.

Should the authors cope with the points listed above, the present manuscript, although quite concise, could become suitable for publication in "Molecules".

Thank-you

Reviewer 3 Report

The authors present a manuscript describing their studies of lewis acid-catalyzed reactions of epoxides with alkenes to form tetrasubstituted tetrahydrofurans. The yields of the reported compounds are low-to-acceptable, but they appear to be characterized sufficiently. The reaction itself, however, has already been reported Hilinski in 2016 (which the authors properly cite as their reference 11). The novelty of the current work, therefore, is extending the reactivity towards tetrasubstituted THFs rather than the 2,4-disubstituted THFs studied in Hilinski’s work.

1. Given the dramatic increase in yield simply by changing from BF2•Et2O to AlC3, I am very surprised that additional Lewis acids were not investigated. Hilinkski found optimal results with Sc(OTf)3. Why was not that Lewis acid at least tested?

2. The scope of the reaction appears to be very limited, with only a 15% yield observed when changing the starting epoxide to something less electron rich (i.e. compound 1c).Essentially, successful results were obtained using only two different starting epoxides (the epoxide leading to 1a, 1b, 1d-1f, and the epoxide leading to 1g). It is very difficult to determine the overall usefulness of such a reaction given such limited studies of an essential component.

3. I would suggest extending the study to other epoxides (for example, does the aromatic ring of the epoxide need to be electron rich? The low yield of 1c suggests that may be the case. If so, is one OCH3 group sufficient? does it matter whether the OCH3 group is para or meta?? Perhaps CH3 substitution is sufficient??)

4. Additional alkenes need to be included in the study. Must they be only 1,2- disubstituted? Will a dialkyl alkene work? How about trisubstitution?

5. In the discussion, I would suggest stating that the configuration assignments that were made based on available magnosalicin are all “tentative”

6. If I understand the syn/anti terminology used by the authors, compound 1a on page 4 should be labelled as anti/anti/syn, should it not?

7. For the mechanism, it is more likely that the reaction is initiated by pre-formation of a cation intermediate formed by reaction of the Lewis acid catalyst (which is listed as BF3 in the mechanism even though that is not the catalyst used!). Likely this is the reason why an electron-donor group is necessary on the aromatic ring of the epoxide: the para substituted electron-donating group on the aromatic ring stabilizes the carbocation sufficiently to allow time for the alkene to react. In the absence of the electron donating group, the Meinwald reaction competes, thereby explaining the low yield of 1c.

Overall, the authors have demonstrated that a very narrow type of THF compound can be synthesized by this method. I would suggest they need to extend their study to convince the reader that it is a synthetic method that can be applied more widely.

Author Response

Dear reviewer:

thanks for your comments. Please find below the responses to your comments:

The authors present a manuscript describing their studies of lewis acid-catalyzed reactions of epoxides with alkenes to form tetrasubstituted tetrahydrofurans. The yields of the reported compounds are low-to-acceptable, but they appear to be characterized sufficiently. The reaction itself, however, has already been reported Hilinski in 2016 (which the authors properly cite as their reference 11). The novelty of the current work, therefore, is extending the reactivity towards tetrasubstituted THFs rather than the 2,4-disubstituted THFs studied in Hilinski’s work.

Given the dramatic increase in yield simply by changing from BF2•Et2O to AlC3, I am very surprised that additional Lewis acids were not investigated. Hilinkski found optimal results with Sc(OTf)3. Why was not that Lewis acid at least tested?

Response: This work was done before the publication of reference 11. We started this work using BF3·Et2O as a catalyst but then changed to cheaper Lewis acid AlCl3 which actually worked much better as it is explained in the manuscript. We were glad to get this result and no other Lewis acids were assayed (a sentence explaining this has been added). This work is a collaboration between our group in Spain (Europe) with the group in Colombia. Colombian authors are experts in isolation of natural products from Colombia and we are dedicated to chemical synthesis. This work was conceived as a convenient procedure for the preparation of natural products including cheap materials also in Colombia. The experiments were done with AlCl3 which is an affordable catalyst everywhere as compared to Sc(OTf)3. For example, the price for AlCl3 in Spain (Merck) is 76.4€/KG which is around one thousand times cheaper than Sc(OTf)3 (84.4€/G).

The scope of the reaction appears to be very limited, with only a 15% yield observed when changing the starting epoxide to something less electron rich (i.e. compound 1c).Essentially, successful results were obtained using only two different starting epoxides (the epoxide leading to 1a, 1b, 1d-1f, and the epoxide leading to 1g). It is very difficult to determine the overall usefulness of such a reaction given such limited studies of an essential component.

Response: The idea of the work was to prepare tetrasubstituted tetrahydrofurans having electron-rich benzenes as the natural products. For that purpose the reaction works. Although the chemical yield is not so high and a mixture of two stereoisomers are obtained, we believe the process is much more efficient than alternative synthetic routes to them. We also assayed the reaction with other alkenes and epoxides but yes then the yield and the stereoselectivity are low. This suggests the mechanism to go through a stabilized carbocation as indicated in Scheme 4.

I would suggest extending the study to other epoxides (for example, does the aromatic ring of the epoxide need to be electron rich? The low yield of 1c suggests that may be the case. If so, is one OCH3 group sufficient? does it matter whether the OCH3 group is para or meta?? Perhaps CH3 substitution is sufficient??)

Response: Regarding to epoxides, isoeugenol and methyl isoeugenol were assayed because they are related to natural products, also styrene oxide with no substitution in the aromatic ring was assayed. Investigation about it is currently underway.

Additional alkenes need to be included in the study. Must they be only 1,2- disubstituted? Will a dialkyl alkene work? How about trisubstitution?

Response: Yes, many assays can be done but it was not the purpose of this work. Thanks for the suggestions. Investigation about it is currently underway.

In the discussion, I would suggest stating that the configuration assignments that were made based on available magnosalicin are all “tentative”

Response: It has been added “tentative stereochemical argument” as suggested by the reviewer. Also a new sentence regarding to this has been added at the end of page 3 (“Interestingly relative 2,4-syn configuration has been also reported for the preparation of disubstituted tetrahydrofurans under similar conditions[11].”)

If I understand the syn/anti terminology used by the authors, compound 1a on page 4 should be labelled as anti/anti/syn, should it not?

Response: The “syn/anti/anti” terminology used has been change to “anti/anti/syn” as suggested by the reviewer.

For the mechanism, it is more likely that the reaction is initiated by pre-formation of a cation intermediate formed by reaction of the Lewis acid catalyst (which is listed as BF3 in the mechanism even though that is not the catalyst used!). Likely this is the reason why an electron-donor group is necessary on the aromatic ring of the epoxide: the para substituted electron-donating group on the aromatic ring stabilizes the carbocation sufficiently to allow time for the alkene to react. In the absence of the electron donating group, the Meinwald reaction competes, thereby explaining the low yield of 1c.

Response: A benzylic cation has been indicated and BF3 has been changed by AlCl3 as suggested by the reviewer.

Overall, the authors have demonstrated that a very narrow type of THF compound can be synthesized by this method. I would suggest they need to extend their study to convince the reader that it is a synthetic method that can be applied more widely.

Response: We agree with the reviewer that the reaction can be extended to more compounds. Investigation about it is currently underway.

Thank-you

Round 2

Reviewer 2 Report

The authors have coped in a satisfactory way with all the issues that I raised in my previous referee report. Consequently, although I still maintain a view similar to referee 3 on the fact that the manuscript is quite concise and could be significantly improved by including the screening of additional Lewis acid catalysts and/or a broader substrate screening, I also think that the manuscript can be considered suitable for publication in Molecules already in its present form.

Author Response

Dear Reviewer:

Thank-you very much. Investigation about new substrates is currently underway.

Sincerely yours.

Reviewer 3 Report

I appreciate the authors attention to my earlier comments/suggestions.  While I still believe the paper offers a very limited scope of products and is likely better as a communication than an article, the limited work presented appears to be well conducted and improved from the first version.  I will leave it up to the editor as to whether to proceed with publication.

1.  Since the authors admit that these reactions are limited to electron rich aryl substituted epoxides, perhaps that should be more clearly communicated in the title of the paper which suggests the reaction is more broad than it is.

2.  I suggested the authors reconsider their proposed mechanism and they did, indeed, make a change, but it is not the change I intended.  Given that the reaction only occurs for electron-rich aryl epoxides, the first step of the reaction is NOT attack by the alkene (as depicted currently in Scheme 4), but ring opening of the Lewis-acid complexed epoxide to form a carbocation.  The carbocation is THEN attacked by the alkene.  We know this to be more likely because if the reaction were initiated by attack of the alkene, the Meinwald products that result from the competing reaction of the epoxide would not be observed (because the Meinwald reaction is intramolecular without incorporation of the alkene into the product).  If, however, ring opening of the epoxide is the first step, then when sufficiently strong electron donor groups are present on the aryl ring to stabilize the carbocation intermediate the intermediate has a sufficient lifetime to react with the alkene.  If, however, there are not sufficiently strong electron donor groups present, the Meinwald reaction competes.

Author Response

Dear Reviewer:

Please find below the responses to your suggestions:

Reviewer:

I appreciate the authors attention to my earlier comments/suggestions.  While I still believe the paper offers a very limited scope of products and is likely better as a communication than an article, the limited work presented appears to be well conducted and improved from the first version.  I will leave it up to the editor as to whether to proceed with publication.

Response: Thank-you.

Since the authors admit that these reactions are limited to electron rich aryl substituted epoxides, perhaps that should be more clearly communicated in the title of the paper which suggests the reaction is more broad than it is.

Response: The title has been changed as suggested.

Reviewer:

I suggested the authors reconsider their proposed mechanism and they did, indeed, make a change, but it is not the change I intended.  Given that the reaction only occurs for electron-rich aryl epoxides, the first step of the reaction is NOT attack by the alkene (as depicted currently in Scheme 4), but ring opening of the Lewis-acid complexed epoxide to form a carbocation.  The carbocation is THEN attacked by the alkene.  We know this to be more likely because if the reaction were initiated by attack of the alkene, the Meinwald products that result from the competing reaction of the epoxide would not be observed (because the Meinwald reaction is intramolecular without incorporation of the alkene into the product).  If, however, ring opening of the epoxide is the first step, then when sufficiently strong electron donor groups are present on the aryl ring to stabilize the carbocation intermediate the intermediate has a sufficient lifetime to react with the alkene.  If, however, there are not sufficiently strong electron donor groups present, the Meinwald reaction competes.

Response: The mechanism has been changed as suggested.

Sincerely yours.